# The Characteristics of Patients That Develop Severe Leptospirosis: A Scoping Review

**DOI:** 10.3390/pathogens14121268

**Published:** 2025-12-10

**Authors:** Patrick Rosengren, Liam Johnston, Ibrahim Ismail, Simon Smith, Josh Hanson

**Affiliations:** 1School of Medicine and Dentistry, James Cook University, Cairns, QLD 4870, Australia; patrick.rosengren@my.jcu.edu.au (P.R.);; 2Department of Medicine, Cairns Hospital, Cairns, QLD 4870, Australia; ibrahim.ismail@health.qld.gov.au (I.I.); simon.smith2@health.qld.gov.au (S.S.); 3The Kirby Institute, University of New South Wales, Kensington, NSW 2052, Australia

**Keywords:** leptospirosis, clinical management, critical care, tropical medicine

## Abstract

This scoping review of original literature published before 1 March 2025 examined the demographic and simple clinical and laboratory findings associated with the development of severe leptospirosis. The definition of severe leptospirosis varied in different studies, but for the purposes of this review it included death or patients with a more complicated clinical course. There were 35 articles that satisfied the review’s inclusion criteria. Increasing age was associated with severe disease in 7 studies. Abnormal respiratory examination findings (18 studies), hypotension (11 studies), oliguria (8 studies), jaundice (7 studies) and altered mental status (4 studies) also helped identify high-risk patients. Abnormal laboratory tests—specifically the complete blood count (17 studies), measures of renal function (16 studies) and liver function (14 studies)—were also associated with severe disease. There was geographical heterogeneity in the clinical phenotype of severe disease, but the presence of hypotension, respiratory or renal involvement had prognostic utility in all regions. Simple bedside findings and basic laboratory tests can provide valuable clinical information in patients with leptospirosis. Integration of these indices into early risk stratification tools may facilitate recognition of the high-risk patient and expedite escalation of care in resource-limited settings where most cases of life-threatening leptospirosis are seen.

## 1. Introduction

Leptospirosis is a globally distributed zoonotic disease which is caused by pathogenic bacterial spirochaetes from the *Leptospira* genus [1]. It was estimated in 2015 that, globally, there were approximately 1 million annual cases of the disease with 60,000 deaths [2]. There is no effective vaccine for human leptospirosis, avoiding exposure to the pathogen is challenging, and chemoprophylaxis is not a practical public health strategy. While most cases of leptospirosis are mild and self-limiting, severe disease can occur with a case fatality rate that can exceed 50% [1,3].

Severe leptospirosis can develop in otherwise well individuals, presenting as a rapidly progressive multisystem illness with clinical manifestations that include shock, respiratory failure, pulmonary haemorrhage and acute kidney injury (AKI). Multimodal critical care support including vasopressors, mechanical ventilation and renal replacement therapy may be required to ensure survival [4]. However, leptospirosis disproportionately affects people living in rural and resource-limited settings, where this sophisticated critical care is often unavailable. In these locations, earlier identification of the patient at highest risk of deterioration can inform clinical management and can expedite transfer to referral centres for delivery of life-saving critical care support.

But of course, the rural and remote healthcare facilities where individuals with leptospirosis first present also frequently lack the diagnostic resources that help identify the high-risk patient. In these locations, where there is often limited access to laboratory and radiology support, clinicians must frequently rely on bedside clinical assessment to determine which patients need prompt referral for escalation of care and which patients can avoid unnecessary and costly transfers.

Researchers from different parts of the world have identified a variety of demographic factors, clinical features and simple laboratory investigations that identify the high-risk patient. However, there are approximately 300 different pathogenic serovars of *Leptospira* and the serovars that are prevalent in different regions vary greatly. This may contribute to a variation in the clinical phenotypes of leptospirosis that are seen in different parts of the world and the clinical signs that have prognostic utility [5,6,7].

This scoping review was performed to define the demographic characteristics, the clinical findings and the simple laboratory investigations that were most associated with a complicated clinical course in patients with leptospirosis in different parts of the world. The review aimed to determine if there were any patterns in the presentation of individuals with more severe disease and if any clinical or laboratory features had universal prognostic utility. It was hoped that this would provide data to inform the optimal clinical care of this unique, life-threatening pathogen in the resource-limited settings where it is most encountered.

## 2. Methods

### 2.1. Scoping Review

This scoping review aimed to identify the demographic, clinical and simple laboratory characteristics of patients presenting with laboratory-confirmed leptospirosis that were most associated with the subsequent development of severe disease. On 2 March 2025, we conducted a search of published literature from five databases, PubMed, Medline (Ovid), Emcare (Ovid), CINAHL (EBSCO) and Scopus, with the search strategy outlined in Appendix A. The search strategy was developed in consultation with a medical librarian with expertise in scoping review methodology. In addition, we examined the references of the articles and other reviews to identify other publications which had not been identified in our initial search. The review was registered with Open Science Framework (http://doi.org/10.17605/OSF.IO/YAGSB) and reported in accordance with the Preferred Reporting Items for Systematic Reviews and Meta-Analysis (PRISMA) guidelines (Appendix A) [8].

### 2.2. Inclusion Criteria

Prospective or retrospective studies of individuals with laboratory-confirmed leptospirosis were eligible for inclusion. For an individual to have a laboratory-confirmed diagnosis, it was necessary to satisfy at least one of the following criteria: (1) *Leptospira* organisms isolated from blood culture; (2) detection of *Leptospira* in blood or urine by polymerase chain reaction (PCR); (3) microscopic agglutination test (MAT) results which satisfied the local definition of confirmed disease. Studies were excluded if there was no comparison between severe and non-severe groups or if no statistically significant findings were discovered; duplicate publications were also excluded.

### 2.3. Study Selection

The searches were completed twice, independently, by two reviewers, P.R. and L.J. Reference management software (Endnote version 9) was utilised to manage references and to identify duplicates. The two reviewers screened all potentially eligible studies independently and aimed to reach agreement about their inclusion. If uncertainty remained, a consensus was achieved during discussions with the senior authors (J.H., I.I., S.S.) until agreement was reached.

### 2.4. Data Extraction and Outcomes

Relevant data from each included study were extracted and collated. These data included the year of publication, the type of study, the geographical location of the study, the number of study participants, the characteristics of the hospital, the definition of severe disease and the factors that the study’s authors identified as being associated with severe disease.

### 2.5. Bias Assessment

The risk of bias for each study was evaluated using the Newcastle–Ottawa Scale (NOS) (Appendix A) [8]. Two reviewers (P.R. and L.J.) independently assessed study quality, considering factors such as study design, participant selection, sample representativeness, method of case verification, and adequacy of control selection and adjustment. Discrepancies in scoring were resolved by discussion until a consensus was reached.

For case–control and cohort studies, the standard NOS criteria were applied, and scores ranged from 0 to 9, with 7–9 stars determined to be low risk of bias, 4–6 stars as being moderate risk of bias and 0–3 stars being high risk of bias. For cross-sectional studies, an adapted version of the NOS was used to ensure appropriate evaluation of study characteristics specific to this design [9]. This adaptation focused on the selection of participants, comparability of study groups, and ascertainment of the outcome (Appendix A).

## 3. Results

We identified 1704 potentially eligible studies; 35 satisfied criteria for inclusion in the review (Figure 1). These 35 studies, stratified by their continent of origin, are summarised in Table 1, Table 2, Table 3 and Table 4 and Figure 2.

### 3.1. Demographic Characteristics

#### 3.1.1. Age

Multiple studies have found that disease severity is correlated with increasing age. Age cut-offs have varied in different reports, but seven studies in our review identified an association between older age and the risk of severe disease [10,11,14,26,27,28,35]. A Thai multicentre study of 2188 participants by Suwannarong et al. found that death and morbidity (jaundice, haemorrhage, cardiac and pulmonary involvement) were 1.3 times more likely in those over 36 years of age [10]. A multicentre retrospective Malaysian study with 525 participants by Al Hariri et al. reported that death was more common in patients greater than 40 years of age [11]. These findings were corroborated by three Brazilian multicentre retrospective studies which found older age (greater than 40 and 60 years in the different series) was associated with an increased risk of death [26,27,28]. A French study of 160 patients requiring ICU admission, with a median age of 54, also found that increasing age was associated with in-hospital mortality [35].

#### 3.1.2. Gender

The incidence of leptospirosis is higher in males than in females; in many studies infection in males is nearly ten times more likely than in females, which is felt to be explained by differences in occupational and recreational exposures [2,45]. However, almost all studies have found no statistically significant difference between males and females in the rates of disease severity or death [10,11,12,18,19,20,23,24,27,29,30,31,35,36,37,43]. In our review, we could identify only one study from the Philippines with 203 participants which reported a significant association between gender and disease severity. In this series, men had an odds ratio (OR) of 3.3 (95% confidence interval (CI): 1.2–12.6) for the composite endpoint of AKI, requirement for dialysis, pulmonary haemorrhage and transaminitis [15].

### 3.2. Clinical Findings

#### 3.2.1. Comorbidities and Lifestyle Factors

Very few papers found an association between comorbidities and severe disease. Two single-centre studies in Guadeloupe and France found that chronic alcohol misuse increased the risk of death [31,35]. In the study from Guadeloupe, a history of hypertension was also a risk factor for death [31]. The Malaysian study by Al Hariri et al. identified that chronic kidney disease was an independent risk factor for deterioration [11]. Meanwhile, a New Caledonian study of 176 patients found that a history of smoking was a risk for death, disease requiring intensive interventions (including renal replacement therapy (RRT), mechanical ventilation, vasopressor support and blood transfusion) or alveolar haemorrhage [43].

#### 3.2.2. Symptoms

A retrospective multicentre Chinese study identified dyspnoea as the strongest independent predictor for ICU admission (OR (95% CI): 19.1 (1.2–692.7)) [18]. A prospective Sri Lankan study of 79 patients also found that dyspnoea at presentation was associated with end organ failure (OR (95% CI): 7.1 (1.3–38.4)) [21]. A 2008 Brazilian study reported that individuals with pulmonary involvement (which included dyspnoea as well as haemoptysis, pulmonary rales on physical examination, and intubation) were 9 times (95% CI 5–17) more likely to die than those without pulmonary involvement [27].

#### 3.2.3. Vital Signs

##### Blood Pressure

Thai, Taiwanese, Malaysian, Sri Lankan, Caribbean, Brazilian and Australian studies have identified an association between hypotension and severe disease [12,13,16,19,20,24,28,29,30,41,43]. A large case–control study of 480 patients from Thailand studies showed that blood pressure < 90/60 mmHg within the first day of admission was associated with death, renal and respiratory insufficiency (OR (95% CI): 17.3 (6.9–43.6)) [12]. A small, retrospective multicentre Taiwanese study of 57 patients determined that blood pressure was the only independent predictor for severity (OR (95% CI):14.8 (3.0–73.6)) of over 50 variables analysed [22].

##### Respiratory Rate

A 2016 Brazilian study of 206 patients found tachypnoea (>25 breaths per minute) was associated with pulmonary haemorrhage (OR (95% CI): 13 (1.3–132)) [29]. Another 2010 Brazilian study which examined respiratory rate as a continuous variable found that a rising rate increased the risk for ICU admission (OR (95% CI): 1.1 (1.1–1.2)) [30]. A study from the French West Indies also found that a respiratory rate over 20 breaths/min increased the risk of death by 11.7 (95% CI 2.8–48.5) [34].

##### Level of Consciousness

The role of altered mental status in predicting severe disease was explored in four studies across the Americas and Europe, but in no study from Asia or Oceania [30,31,35,39]. A French retrospective study conducted with 160 patients admitted to ICU found confusion as a predictor for death [35]. A Turkish study found that altered mental status raised risk of subsequent death by 8.9 (95% CI: 1.6–50.7) [39]. A Brazilian study with a focus on pulmonary haemorrhage determined that a Glasgow Coma Score (GCS) < 15 was associated with a 7.7 (95% CI 1.3–23.0) times the risk of pulmonary haemorrhage and, after shock at presentation, was the most predictive factor of this complication [30]. Finally, a study from Guadeloupe found that impaired consciousness was associated with death, requirement for dialysis or mechanical ventilation (OR (95% CI): 3.8 (1.1–13.2)) [31].

##### Heart Rate, Temperature and Pulse Oximetry

Our search strategy did not find any papers examining the independent prognostic value of heart rate, temperature or pulse oximetry.

#### 3.2.4. Bedside Examination Findings

##### Jaundice

Jaundice was identified as a predictive factor for severe disease in seven studies from Brazil, France and the French West Indies, Sri Lanka and Thailand [12,21,25,31,32,35,40]. A small, single-centre French cohort study of 62 patients found that patients with jaundice at any point during admission were 10.1 times (95% CI 1.8–56.8) more likely to require ICU admission or RRT than patients without jaundice [40]. In another French study of patients admitted to ICU, the risk of death was greater in those with than without jaundice [35]. A retrospective Thai study of 480 patients showed the presence of jaundice at admission raised the likelihood of death and renal and respiratory failure threefold (95% CI: 1.7–5.7) [12].

##### Respiratory Auscultation Abnormalities

Studies from Australia, Guadeloupe, Malaysia, Martinique and Thailand have reported that patients presenting with any abnormality on respiratory auscultation have more severe disease than those without changes on auscultation [13,16,31,32,41]. Most of these studies included any changes detected on auscultation in their definition; only one prospective Thai study of 121 patients reported pulmonary rales specifically as a risk for death (RR (95% CI): 5.2 (1.4–19.9)) [16].

##### Oliguria

Reports from Australia, Brazil, the French West Indies, Sri Lanka, Thailand and Ukraine have reported that oliguria (defined in different studies as a daily urine output below 400 or 500 mL) is associated with severe disease and death [16,21,27,31,32,34,37,41,46]. Authors of a retrospective Australian study of 402 adults found that oliguria had the greatest independent prognostic utility of any of the clinical variables they analysed (OR (95% CI): 16.4 (6.9–38.8)) [41]. A French West Indies group found that patients with oliguria at presentation had an OR (95% CI) of 9.0 (2.1–37.9) for death [34].

### 3.3. Laboratory Findings

#### 3.3.1. Haematology

An Australian study found that anaemia, leucocytosis and thrombocytopenia were all associated with ICU admission [41]. A Sri Lankan study also showed that anaemia, leucocytosis and thrombocytopaenia were associated with severe disease, but this study also showed that leucopoenia was associated with disease severity [14,42]. A Ukrainian case–control study which found leucocytosis and thrombocytopaenia were associated with disease severity, as did studies from the Philippines and Puerto Rico [15,33,37].

A Chinese retrospective multicentre study of 95 patients discovered a neutrophilia to be the only laboratory variable that independently predicted ICU admission (OR (95% CI):1.6 (1.0–2.5)) [18]. Similarly, a Sri Lankan study identified that not only leucocytosis >11,000 mm^3^ (OR (95% CI): 3.6 (1.3–9.9) but also a neutrophil percentage of >75% (OR (95% CI):13.4 (1.7–108.1)) correlated with disease severity [21].

Studies in Brazil, Malaysia and Thailand identified that a lower haematocrit was associated with severe disease [12,13,28]. The Brazilian study found that when haematocrit was below 30%, the risk of deterioration increased by a factor of 3.5 (95% CI 1.6–7.6) [28].

Studies in Malaysia, Martinique and Türkiye all found a link between a prolonged prothrombin time (PT) and disease severity [11,32,39]. In the study from Türkiye, 66.7% of the patients who died had a prolonged PT compared to only 22.2% of survivors. In the Malaysian study by Al Hariri et al., both PT and activated partial thromboplastin time (aPTT) were associated with disease severity; however it showed that a prolonged PT was more frequent in the cohort (179/525 (34%) had a prolonged PT compared to 40/525 (7.6%) who had a prolonged aPTT) [11,39].

#### 3.3.2. Biochemistry

##### Urea and Creatinine

An elevated serum creatinine at presentation was associated with severe disease in studies from Australia, Brazil, Puerto Rico, and Ukraine [21,27,30,33,37,41,42]. Acute Kidney Injury (AKI, variably defined across studies) was seen in multiple studies across Brazil and Malaysia as a predictor of ICU admission and death, with the Brazilian cohort study finding it raised the risk of ICU admission 14-fold (95% CI 1.3–150) [20,29]. The multicentre Malaysian study of 525 patients by Al Hariri et al. found that over 80% of the fatal cases had AKI compared to only 42.8% of the survivors [11]. Multiple studies found that an elevated creatinine was associated with a complicated disease course, like a New Caledonian study which found a serum creatinine > 200 µmol/L raised likelihood of death or requirement for intensive medical intervention by 5.86 times (95% CI 1.6–21.3) [43]. This was echoed in a Sri Lankan study which found that an elevated serum creatinine >120 μmol/L in a small cohort of 79 patients was the strongest predictor of severe disease with an OR of 29.1 (95% CI 6.1–140.2) [21].

##### Electrolytes

A Sri Lankan study found that a serum sodium < 131 mEq/L at presentation independently predicted severe disease (OR (95% CI): 6.4 (1.4–30.4)) [14]. A Brazilian study showed an association between hyperkalaemia and pulmonary haemorrhage, whilst a prospective cohort study in Thailand found that hyperkalaemia was independently associated with death (RR (95% CI): 5.9 (1.7–21)) [16,30].

##### Bilirubin

Studies from Sri Lanka, Ukraine and from France and its overseas territories have each identified a correlation between bilirubin levels at admission and the risk of subsequent deterioration [21,35,36,37]. A single-centre New Caledonian study of 47 patients found bilirubin >35 µmol/L increased the risk of death and need for dialysis fivefold (95% CI 1.3–20) [44].

##### Other Liver Function Tests

A Malaysian study found that a serum ALT > 50 IU/L increased the likelihood of a patient having AKI or pulmonary involvement (OR (95% CI): 2.9 (1.2–7.2)) [20]. A Sri Lankan study identified that an ALT > 70 IU/L at presentation was the strongest risk factor for death, ICU admission, and prolonged hospital stay [14]. Another 2023 Sri Lankan study that examined 88 patients found that an elevated AST was associated with subsequent death (OR (95% CI): 5.3 (1.6–17.1)) [19]. A multicentre retrospective Indian study of 101 patients found that the AST:ALT ratio was the only laboratory finding to predict severe disease [17].

### 3.4. Other Investigations

#### 3.4.1. Chest Radiological Findings

One study of 68 patients in the French West Indies found that the presence of alveolar infiltrates on chest X-ray at presentation increased the risk of death by 7.3 times (95% CI 1.7–31.7) [34].

#### 3.4.2. Electrocardiogram Changes

A retrospective Malaysian study found that non-specific T wave changes, conduction abnormalities and atrial fibrillation (AF) on electrocardiogram (ECG) were significantly associated with death [11]. A prospective Sri Lankan study of 88 patients found that AF was the only ECG abnormality that was associated with death [19]. A retrospective multicentre study from the French West Indies reported that ECG repolarization abnormalities predicted death [21,34]. Finally, a French group examined a cohort of 62 patients and looked for ECG abnormalities and clinical and laboratory evidence of cardiac involvement (including shock, myocarditis or pericarditis), discovering that the presentation of this cardiac involvement increased the risk of ICU admission or dialysis by 31.2 times (95% CI 1.8–550) [40].

## 4. Discussion

This review illustrates that bedside findings and simple laboratory tests can rapidly identify patients with leptospirosis who are at greatest risk of a more complicated disease course. Although the studies were performed in different continents and examined a diverse range of variables in both resource-rich and resource-limited settings, it was notable that patients with hypotension, signs of respiratory involvement or evidence of renal involvement were consistently identified as the patients who were at greatest risk of deterioration and death. The presence of these findings should, in the appropriate clinical context, encourage clinicians to consider escalation of care or, in resource-limited settings, transfer to a referral centre. Their presence can also inform the supportive care that these patients receive during their hospitalisation with this life-threatening disease.

The presence of respiratory involvement predicted severe disease in 18 of the studies, emphasising its reputation as perhaps the most feared manifestation of leptospirosis [19,47]. The definition of respiratory involvement varied in the different studies; although the presence of abnormal findings on auscultation was examined most commonly, tachypnoea and changes on plain chest X-ray also had prognostic utility. The manifestations of pulmonary leptospirosis range from mild dyspnoea and cough to life-threatening pulmonary haemorrhage and acute respiratory distress syndrome (ARDS) [48]. These clinical findings reflect the underlying pulmonary oedema and haemorrhage that are seen in patients with lung involvement. Although the pathophysiology of respiratory involvement is incompletely understood, it is believed to result from increased permeability due to vascular injury and impaired fluid handling [49,50,51]. The presence of lung involvement may not only suggest that a patient is at risk of, sometimes rapid, deterioration but it should also discourage clinicians from delivering aggressive fluid resuscitation that might exacerbate incipient pulmonary oedema [52]. We did not find any studies examining the utility of pulse oximetry in the management of patients with leptospirosis, although intuitively it would appear to be well suited to the resource-limited settings where most cases of leptospirosis are seen [2,53,54]. This may be a focus for future studies.

AKI is another classic manifestation of severe leptospirosis that is associated with significant morbidity and mortality [14,55,56]. Laboratory evidence of renal involvement—most commonly an elevated serum creatinine—had prognostic utility in 16 studies. Meanwhile the bedside finding of oliguria—more suitable to settings where there is limited laboratory support—predicted a more complicated disease course in 8 studies. It is important to recognise AKI promptly as it is life-threatening and can evolve rapidly, but it can be successfully treated with—an often short course of—RRT, which may require early transfer to a referral centre [1,36,57,58]. Hypotension—one of the five so-called “vital signs” which can be measured at presentation and followed sequentially—was also frequently of prognostic value. This also has a simple remedy in the form of cautious fluid resuscitation and early vasopressor support [4,35,36]. If hypotension is recognised and addressed promptly, adverse sequelae can be minimised [59].

For many clinicians, the perception persists that severe leptospirosis is characterised by jaundice; indeed severe leptospirosis is commonly referred to as icteric leptospirosis (or Weil’s disease) in the literature [1,60]. Hepatic involvement (jaundice, deranged liver function tests or hepatomegaly) was a predictor of severe disease in every European study in our review; however, it is assuredly the case that patients with life-threatening leptospirosis are not always icteric [6,61]. In Australia, for instance, less than 10% of patients with severe leptospirosis have jaundice [41,42,57]. The explanation for this apparent geographical heterogeneity in the clinical phenotype of severe leptospirosis is likely to be multifactorial, but it may be at least partly explained by variation in the locally prevalent serovars. Infection with *Leptospira interrogans* serogroup Icterohaemorrhagiae was associated with severe disease in studies from Martinique (where 75% of the patients with severe disease had jaundice) and New Caledonia (where over 85% of patients with severe disease had jaundice) [32,43]. In contrast, serovars from the Pyrogenes, Australis and Sejroe serogroups are identified most commonly in tropical Australia where jaundice is uncommon in patients with severe disease [57,62].

But it is important to note that icteric patients with leptospirosis also frequently have hypotension, respiratory involvement and AKI, and it is these complications—not the hepatic dysfunction—that are most commonly responsible for patient death [16,30,32,43,47,63]. These three variables have been incorporated into the SPiRO score, a simple disease severity score that has been devised in Australia [41]. One of the advantages of the SPiRO score is that its three elements—systolic blood pressure < 100 mmHg, abnormal auscultatory findings on respiratory examination and oliguria—can be determined at the bed side in resource-limited settings by even junior health workers in less than a minute. An absence of any of the three clinical variables of the SPiRO score at presentation had a negative predictive value for the subsequent development of severe disease of 97%. The score was derived in tropical Australia’s well-resourced heath system and requires validation in other settings but our review suggests that it may have greater global utility—and be easier to use in clinical practice—than the Brazilian QuickLepto score, another score devised to predict death in leptospirosis patients at hospital admission [28]. The QuickLepto score has prognostic utility, but has five variables and requires the haematocrit which may not be accessible as rapidly in the locations where leptospirosis is most commonly seen [2,28].

However, an important caveat to consider when assessing the clinical utility of the SPiRO and the QuickLepto scores is that they were derived from cohorts of patients with an established diagnosis of leptospirosis [1]. While epidemiological and clinical findings may lead health workers to suspect leptospirosis in a patient—and in many high-incidence settings the majority of patients will have leptospirosis suspected at presentation—the actual diagnosis is often not confirmed for several days, even in high-income settings [1,57]. The clinical and laboratory findings that are suggestive of leptospirosis can also be seen in patients with many other infections encountered in rural and remote tropical settings, including dengue, malaria, Q fever and rickettsial disease, and it is unwise to use a prediction score developed for patients with leptospirosis in a patient who may have a quite different condition [64,65,66,67]. This limits the practical applicability of the leptospirosis-specific scores which, perhaps relevantly, have not been shown to be better at identifying patients at risk of deterioration than commonly used prediction scores such as the National Early Warning Score 2 (NEWS2), the Systemic Inflammatory Response Syndrome (SIRS) score and the Universal Vital Assessment (UVA) score [68,69,70,71,72]. As hypotension and respiratory involvement are associated with poorer outcomes in patients with leptospirosis, it might be anticipated that these validated, simple disease severity scores will also assist the triage of patients with leptospirosis, well before their diagnosis is established. The relative utility of commonly used general scores (like NEWS2, SIRS and UVA) and leptospirosis-specific scores (like SPiRO, THAI-LEPTO and QuickLepto) in resource-limited settings where the diagnosis of leptospirosis is not confirmed should be a focus of future studies [24].

Our study has many limitations. While we did our best to identify as many relevant studies as possible, it is likely that we have missed some, particularly those that were not published in English. The definition of severe disease—and the prognostic factors that were examined in each of the reports—was not standardised, although we do not feel that this would necessarily change the main findings of our report. This methodological flexibility also allowed us to compare cohorts managed in different health systems where there may be differences in access to care and the advanced critical care support that sometimes may be necessary for survival. While cardiac involvement and altered consciousness were not identified as common predictors of severe disease, both of these findings are associated with a higher risk of mortality and should always be sought in a patient in whom leptospirosis is a possible diagnosis [7,73]. As ever, patient management should always be informed by thoughtful and repeated clinical assessment and tailored to the individual patient and the health system in which they are managed [74]. Our report had a biomedical focus and did not examine the social determinants of health and their role in disease outcomes, although they undoubtedly contribute [75,76,77,78]. We also did not examine pathogen-related factors such as the infecting serovar, although it is only possible to determine this in regions with access to advanced laboratory support and this information is not available in a timely enough manner to inform patient triage and clinical care, which were the focus of this study [1,48]. Importantly, serologic analysis—which is most commonly used to infer the infecting serovar—has poor sensitivity and specificity for the identification of the infecting serovar in individual cases of leptospirosis in humans and have not been shown to have a role in the management of patients with the disease [65,79].

In future prospective studies, it will be important to define the relative contributions of host, pathogen and environmental factors to the clinical phenotype of leptospirosis. This will also provide important insights into the pathophysiology of severe leptospirosis and the utility of different adjunctive therapies. It will be important to define, in particular, the independent contribution of the responsible serovar for the patients’ clinical phenotype and how much this explains the variation in the clinical presentation that exists across regions [80]. There is growing interest in the prognostic utility of quantitative PCR in leptospirosis as well as a range of biomarkers including plasma neutrophil gelatinase-associated lipocalin (NGAL) and long pentraxins [32,43,81,82]. The relationship between these variables, clinical phenotypes and patient outcomes will also provide greater insights into the pathophysiology and the optimal therapy of severe leptospirosis which remains incompletely defined [83,84,85,86,87]. More practically, it is also important to determine the independent contribution of access to quality care and time to presentation on disease severity and outcomes [15,17,31,43]. Finally, it remains to be seen whether the identification and management of the high-risk patient with leptospirosis is substantially different to the management of patients with other serious infections [59,88].

## 5. Conclusions

A range of simple clinical and laboratory indices can identify the patient with leptospirosis who is at greatest risk of deterioration. Although there is some geographical variation in the clinical phenotype of severe leptospirosis, the presence of hypotension, respiratory involvement and renal involvement have been repeatedly associated with severe disease across diverse populations. Thorough assessment of patients to identify these findings in patients with suspected leptospirosis should expedite recognition of the patients at greatest risk of deterioration and encourage the clinician to consider escalation of their care. Importantly, these manifestations can be identified rapidly and inexpensively in the resource-limited settings where patients with leptospirosis are most commonly seen, enabling the prompt delivery of optimal supportive care for this life-threatening infection.

## Figures and Tables

**Figure 1 pathogens-14-01268-f001:**
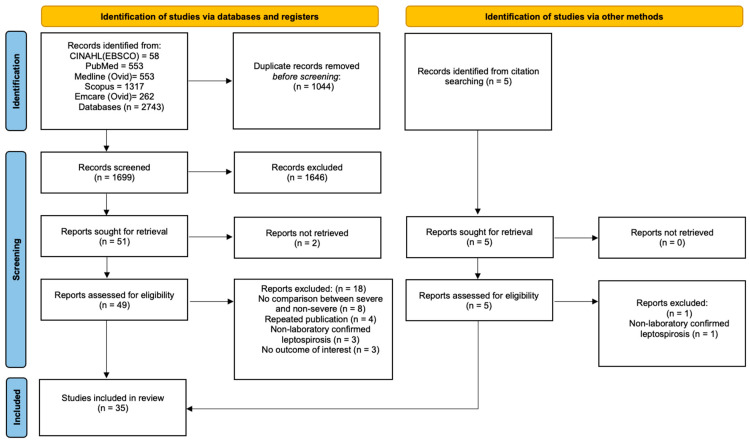
PRISMA flow diagram of the study selection process. CINAHL, cumulative index of nursing and allied health; EBSCO, Elton B. Stephens Company.

**Figure 2 pathogens-14-01268-f002:**
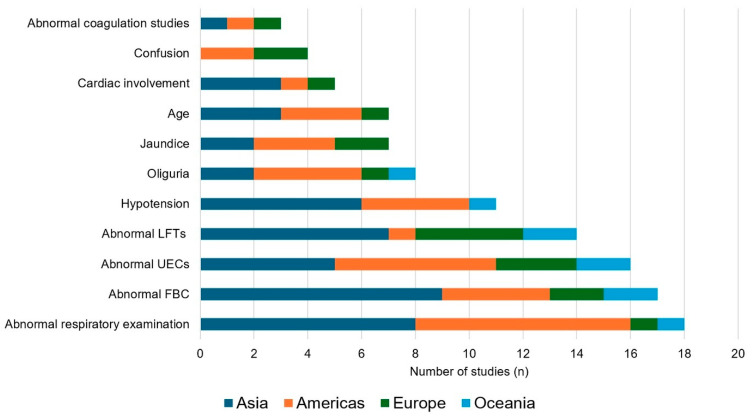
The association of selected variables with severe Leptospirosis, stratified by global region. UECs: Urea electrolytes and creatinine; FBC: Full blood count; LFTs: Liver function tests.

**Table 1 pathogens-14-01268-t001:** Summary of the findings of the 15 studies from Asia that were included in this scoping review.

Article	Location	Number of Patients	Primary or Referral Hospital	Type of Study	Definition of Severe Disease	Variables Associated with Severe Disease	Odds Ratio	95% Confidence Interval
Suwannarong et al., 2014 [10]	Thailand	2188	Both	Retrospective multicentre	Death OR Renal dysfunction OR Jaundice OR Haemorrhagic manifestations OR Leucocytosis OR Cardiac involvement OR Pulmonary involvement	Age > 36 years	1.3	1.1–1.5
						Residence in rural area	0.7	0.5–0.96
						Late initiation of treatment (>2 days vs. ≤2 days after symptom onset)	5.4	3.1–9.3
Al Hariri et al., 2022 [11]	Malaysia	525	Referral	Retrospective multicentre	Death	Age > 40 years	Not reported	-
						CKD	Not reported	-
						Tachypnoea (RR > 28/min)	Not reported	-
						AKI	Not reported	-
						Rhabdomyolysis	Not reported	-
						Multiple organ dysfunction	Not reported	-
						Respiratory failure	Not reported	-
						Pneumonia	Not reported	-
						Sepsis	Not reported	-
						T-wave changes on ECG	Not reported	-
						Atrial fibrillation	Not reported	-
						Conducting abnormality	Not reported	-
						Venous acidosis	Not reported	-
						Elevated AST or ALT	Not reported	-
						Hypoalbuminemia	Not reported	-
						Severe thrombocytopenia	Not reported	-
						Prolonged PT	Not reported	-
						Prolonged aPTT	Not reported	-
						Pulmonary infiltrate	Not reported	-
Pongpan et al., 2023 [12]	Thailand	480	Primary	Retrospective multicentre	Death OR Serum creatinine > 3 mg/dL OR Respiratory failure	Haemoptysis	25.8	5.7–116.9
						Hypotension (BP < 90/60 mmHg)	17.3	6.9–43.6
						Jaundice	3.1	1.7–5.7
						Thrombocytopaenia < 100,000/µL	8.4	4.7–15.1
						Leucocytosis > 14,000/µL	5.1	2.8–9.5
						Haematocrit ≤ 30%	3.5	1.6–7.6
Sandhu et al., 2020 [13]	Malaysia	456	Both	Retrospective multicentre	Jaundice OR Renal dysfunction OR Haemorrhaging OR Myocarditis OR Arrhythmia OR Pulmonary haemorrhage with respiratory failure OR Meningitis/Meningoencephalitis	Abnormal respiratory auscultation	3.1	1.6–6.0
						Hypotension	2.2	1.1–4.3
						Hepatomegaly	7.1	1.1–46.0
						Leucocytosis	2.1	1.4–3.3
						Low haematocrit	2.3	1.4–3.8
						Increased ALT	2.1	1.4–3.3
Rajapakse et al., 2015 [14]	Sri Lanka	232	Referral	Prospective multicentre	Death OR ICU admission OR Hospital stay > 10 days OR Evidence of major organ dysfunction (liver, kidney, lung or heart)	Age > 40 years	Not reported	
						Highest recorded fever > 38.8 °C	Not reported	
						Myalgia	Not reported	
						PCV < 29.8%	3.8	1.4–10.4
						PCV > 33.8	Not reported	
						Hb < 10.2 g/dL and >11.2 g/dL	Not reported	
						ALT > 70 IU/L	2.6	1.03–6.8
						Hyponatremia < 131 mEq/L	6.4	1.4–30.4
						WBC > 12 350/mm^3^ and <7900 mm^3^	Not reported	
						Thrombocytopaenia < 63 500/mm^3^	Not reported	
Lee et al., 2017 [15]	Philippines	203	Referral	Prospective single centre	AKI OR Dialysis OR Pulmonary haemorrhage OR Liver dysfunction (2.5× ULN AST and ALT or presenting with jaundice).	Male sex	3.3	1.2–12.6
						Duration of symptoms prior to antibiotic therapy	1.3	1.1–1.5
					Death	Neutrophilia	1.38	1.2–1.7
						Thrombocytopenia	0.99	0.97–0.99
Panaphut et al., 2002 [16]	Thailand	121	Referral	Prospective single centre	Death	Hypotension	RR = 10.3	1.3–83.2
						Oliguria	RR = 8.8	2.4–31.8
						Hyperkalaemia	RR = 5.9	1.7–21.0
						Pulmonary rales on auscultation	RR = 5.2	1.4–19.9
Goswami et al., 2014 [17]	India	101	Referral	Retrospective multicentre	Death	Duration of symptoms prior to antibiotic therapy	1.3	1.1–1.6
						AST/ALT ratio	1.2	1.1–1.4
Li et al., 2022 [18]	China	95	Referral	Retrospective multicentre	ICU admission	Dyspnoea	29.1	1.2–692.7
						Neutrophilia	1.6	1.03–2.5
Fonseka et al., 2023 [19]	Sri Lanka	88	Referral	Prospective single centre	Death	Pulmonary haemorrhage	9.3	2.4–36.1
						Hypotension (BP < 90/60)	12.2	1.5–97.7
						Atrial fibrillation	4.7	1.2–17.8
						Acute haemoglobin reduction	5.3	1.5–18.9
						High AST level	5.3	1.6–17.1
Philip et al., 2021 [20]	Malaysia	83	Referral	Prospective multicentre	Hospitalisation AND Jaundice OR AKI OR pulmonary involvement	AKI	10.4	1.1–100.6
						ALT > 50 IU	8.1	1.2–52.5
						Thrombocytopaenia < 150 × 10^9^/L	7.3	1.5–33.9
Nisansala et al., 2023 [21]	Sri Lanka	79	Referral	Prospective multicentre	Acute kidney injury OR Pulmonary haemorrhage OR Myocarditis OR Liver failure	Dyspnoea	7.1	1.3–38.4
						Icterus	6.5	1.7–24.1
						Oliguria	5.2	1.9–14.6
						Cardiac arrythmias	5.9	1.1–33.2
						WBC > 11,000 mm^3^	3.6	1.3–9.9
						Neutrophils > 75%	13.4	1.7–108.1
						SGOT > 40 U/L	5.9	1.2–28.5
						Serum creatinine > 120 μmol/L	29.1	6.1–140.2
						Blood urea > 6.5 mmol/L	15.0	1.8–123.6
						Total bilirubin > 21 μmol/L	15.2	3.5–66.3
Wang et al., 2020 [22]	Taiwan	57	Referral	Retrospective multicentre	RRT OR Mechanical ventilation OR Vasopressors OR Blood transfusion OR Meningitis or meningoencephalitis OR Myocarditis	Shock	14.8	3.0–73.6
					Death	Previous corticosteroid use	20.2	1.9–217.5
						Haemorrhage	71.2	4.9–999.9
Budiono et al., 2009 [23]	Indonesia	55	Referral	Retrospective single-centre	Death	Pulmonary involvement	9.9	1.2–84.0
						Meningismus	Not reported	
Ajjimarungsi et al., 2020 [24]	Thailand	46	Referral	Retrospective single-centre	ICU admission	SOFA score > 6	1.8	1.2–2.7
						THAI-LEPTO score > 6	1.3	1.1–1.7
						Mechanical ventilator support	63.2	5.8–691.4
						Inotrope or vasopressor requirements	53.5	5.5–524.5

ICU: Intensive care unit; RRT: Renal replacement therapy; AST: Aspartate Aminotransferase; ALT: Alanine transaminase; SGOT: Serum glutamic oxaloacetic transaminase; IU: International units; SOFA: Sequential organ failure assessment; Hb: Haemoglobin; ULN: upper limit of normal; WBC: White blood cell; PCV: Packed Cell Volume; aPTT: activated partial thromboplastin time; PT: Prothrombin Time; CKD: Chronic kidney Disease; AKI: Acute Kidney Injury.

**Table 2 pathogens-14-01268-t002:** Summary of the findings of the 10 studies from the Americas that were included in this scoping review.

Article	Location	Number of Patients	Primary or Referral Hospital	Type of Study	Definition of Severe Disease	Variables Associated with Severe Disease	Odds Ratio	95% Confidence Interval
Silva et al., 2024 [25]	Brazil	1319	Both	Retrospective multicentre	Death	Delay in medical attention	Not reported	
						Headache	Not reported	
						Calf pain	Not reported	
						Vomiting	Not reported	
						Jaundice	Not reported	
						Renal insufficiency	Not reported	
						Respiratory alterations	Not reported	
Daher Ede et al., 2019 [26]	Brazil	507	Referral	Retrospective multicentre	Death OR RRT	Age > 60 years	Death: 3.5 RRT: 2.0	Death: 1.9–6.4 RRT: 1.2–3.5
Spichler et al., 2008 [27]	Brazil	378	Both	Retrospective multicentre	Death	Age > 40 years	2.4	1.4–4.0
						Oliguria	7.1	3.6–15.0
						Pulmonary involvement	9.1	5.0–17.0
						Thrombocytopenia < 70,000/µL,	2.6	1.5–5.0
						Creatinine > 3 mg/dL	4.2	2.4–7.2
Galdino et al., 2023 [28]	Brazil	295	Referral	Retrospective multicentre	Death	Age > 40 years	Not reported	
						Lethargy	Not reported	
						Respiratory symptoms	Not reported	
						Mean Arterial Pressure < 80 mmHg	Not reported	
						Haematocrit < 30%	Not reported	
Daher Ede et al., 2016 [29]	Brazil	206	Referral	Retrospective single centre	ICU admission	Tachypnoea	13.0	1.3–132.0
						Hypotension	5.3	1.5–18.0
						AKI	14.0	1.3–150.0
Marotto et al., 2010 [30]	Brazil	203	Referral	Retrospective single centre	Pulmonary haemorrhage	Respiratory rate	1.1	1.1–1.2
						Presenting in shock	20.1	20.1–236.4
						Glasgow Coma Scale Score < 15	7.7	1.3–23.0
						Hyperkalaemia	2.6	1.1–5.9
						Serum creatinine	1.2	1.1–1.4
Herrmann-Storck et al., 2010 [31]	Guadeloupe	168	Referral	Retrospective single centre	Death OR RRT OR Mechanical ventilation	Chronic hypertension	30.9	6.0–157.4
						Chronic alcoholism	16.8	4.1–57.9
						Duration of symptoms prior to antibiotic therapy	4.8	1.1–20.2
						Abnormal respiratory auscultation	8.7	1.8–41.3
						Jaundice	5.9	1.1–31.1
						Oliguria	5.6	1.5–20.6
						Altered consciousness	3.8	1.1–13.2
						AST > 102 IU/L	4.3	1.2–14.6
						Amylase > 285 IU/L	18.5	3.8–88.8
						*Leptospira interrogans* serovar Icterohemorrhagiae	5.3	1.0–26.0
Hochedez et al., 2015 [32]	Martinique	102	Referral	Retrospective single centre	Death OR Vasopressors OR Dialysis OR Mechanical ventilation OR Blood transfusion	Hypotension	Not reported	-
						Chest auscultation abnormalities	Not reported	-
						Icterus	Not reported	-
						Oligo/anuria	Not reported	-
						Thrombocytopenia	Not reported	-
						PT < 68%	Not reported	-
						High levels of leptospiremia	Not reported	-
						*L. interrogans* serovar Icterohemorrhagiae	Not reported	-
						*L. interrogans* serovar Copenhageni	Not reported	-
Sharp et al., 2016 [33]	Puerto Rico	73	Referral (controls did not have leptospirosis)	Retrospective multicentre	Death	Decreased serum bicarbonate	Not reported	-
						Elevated serum creatinine	Not reported	-
						Leucocytosis	Not reported	-
						Thrombocytopenia	Not reported	-
Dupont et al., 1997 [34]	French West indies	68	Referral	Retrospective single centre	Death	Oliguria	9.0	2.1–37.9
						Dyspnoea	11.7	2.8–48.5
						Leucocytosis > 12,900/mm^3^	2.5	1.8–3.5
						Repolarization abnormalities on ECG	5.9	1.4–24.8
						Alveolar infiltrates on chest x-ray	7.3	1.7–31.7

ICU: Intensive care unit; RRT: Renal replacement therapy; AST: Aspartate Aminotransferase; ALT: Alanine transaminase; SOFA: Sequential organ failure assessment; Hb: Haemoglobin; WBC: White blood cell; PCV: Packed Cell Volume; PT: Prothrombin Time; CKD: Chronic Kidney Disease; AKI: Acute Kidney Injury; GCS: Glasgow come scale.

**Table 3 pathogens-14-01268-t003:** Summary of the findings of the 6 studies from Europe that were included in this scoping review.

Article	Location	Number of Patients	Primary or Referral Hospital	Type of Study	Definition of Severe Disease	Variables Associated with Severe Disease	Odds Ratio	95% Confidence Interval
Miailhe et al., 2019 [35]	France	160	Referral	Retrospective multicentre	Death	Increasing age	Not reported	-
						Chronic alcohol abuse	Not reported	-
						High SOFA score	Not reported	-
						Need for invasive ventilation or RRT within 48 h after ICU admission	Not reported	-
						Jaundice	Not reported	-
						Confusion	Not reported	-
						Higher blood bilirubin level	Not reported	-
						Leucocytosis	Not reported	-
Delmas et al., 2018 [36]	Réunion island (France)	134	Referral	Prospective single centre	Death	Mechanical Ventilation	Not reported	-
						Vasopressors or inotropic support	Not reported	-
						Neurologic and respiratory impairment	Not reported	-
						Transfusion	Not reported	-
						SOFA score	Not reported	-
						SAPS II	Not reported	-
						Acidosis	Not reported	-
						Lower base excess	Not reported	-
						Lactatemia	Not reported	-
						Hyperkalaemia	Not reported	-
						Total bilirubin at admission	Not reported	-
						Intra-alveolar haemorrhage in the first 7 days in ICU	Not reported	-
Petakh et al., 2022 [37]	Ukraine	102	Referral	Retrospective single centre	Death	Oliguria	13.5	2.6–71.1
						Elevated serum creatinine	Not reported	-
						Elevated serum urea	Not reported	-
						Direct and total bilirubin	Not reported	-
						Thrombocytopenia	Not reported	-
						Leucocytosis	Not reported	-
Gancheva et al., 2016 [38]	Bulgaria	100	Referral	Retrospective single centre	Severe indication OR jaundice with severe hepatic dysfunction OR skin haemorrhages and visceral bleeding OR myocarditis OR dialysis OR respiratory and CNS involvement	Age	Not reported	-
Esen et al., 2004 [39] ^a^	Türkiye	72	Referral	Retrospective single centre	Death	Altered mental status	8.9	1.6–50.7
						Hepatomegaly	Not reported	-
						Haemorrhage	Not reported	-
						Increased AST + ALT	Not reported	-
						Prolonged PT	Not reported	-
						Hyperkalaemia	4.2	1.4–13.1
Abgueguen et al., 2008 [40]	France	62	Referral	Retrospective single centre	ICU admission OR RRT	Jaundice	10.1	1.79–56.8
						Cardiac damage (clinical or ECG)	31.2	1.76–50.0

ICU: Intensive care unit; RRT: Renal replacement therapy; AST: Aspartate aminotransferase: ALT: Alanine transaminase; SOFA: Sequential organ failure assessment; SAPS II: Simplified acute physiology score II; PT: Prothrombin time; CKD: Chronic kidney disease; AKI: Acute kidney injury; GCS: Glasgow coma scale; ECG: electrocardiogram. ^a^ Although the majority of the landmass of Türkiye lies within Asia, the study by Esen et al. is presented with the European studies as it was performed in Samsun, Türkiye which is geographically far closer to the European studies in this table than the Asian studies presented in Table 1.

**Table 4 pathogens-14-01268-t004:** Summary of the findings of the 4 studies from Oceania that were included in this scoping review.

Article	Location	Number of Patients	Primary or Referral Hospital	Type of Study	Definition of Severe Disease	Variables Associated with Severe Disease	Odds Ratio	95% Confidence Interval
Smith et al., 2019 [41]	Australia	402	Both	Retrospective multicentre	ICU admission OR RRT OR Mechanical ventilation OR Vasopressor support OR Pulmonary haemorrhage	Oliguria	16.4	6.9–38.8
						Hypotension	4.3	1.7–10.7
						Abnormal respiratory auscultation	11.2	4.7–26.5
Craig et al., 2009 [42]	Australia	239	Both	Retrospective multicentre	ICU admission	Elevated serum urea	Not reported	
						Elevated serum creatinine	Not reported	
						Hypoalbuminemia	Not reported	
						Haematocrit	Not reported	
						Anaemia	Not reported	
						Thrombocytopenia	Not reported	
						Leucocytosis	Not reported	
Tubiana et al., 2013 [43]	New Caledonia	176	Referral	Retrospective multicentre	Death OR RRT OR Mechanical ventilation OR Vasopressor support OR Alveolar haemorrhage OR Blood transfusion	Current cigarette smoking	2.9	1.5–6.0
						Delay > 2 days between the onset of symptoms and the initiation of antibiotic therapy	2.8	1.3–5.9
						Thrombocytopenia ≤ 50,000/µL	6.4	1.8–22.6
						Serum creatinine > 200 µmol/L	5.9	1.6–21.3
						Serum lactate > 2.5 mmol/L	5.1	1.6–16.9
						Serum amylase > 250 IU/L	4.7	1.4–15.7
						>1000 leptospires/mL	4.3	1.2–15.9
						Icterohaemorrhagiae serovar	2.8	1.3–6.2
Mikulski et al., 2014 [44]	New Caledonia	47	Referral	Prospective single centre	Death OR Mechanical ventilation OR Dialysis	LDH ≥ 390 IU/L	5.8	1.3–25.6
						Increased total bilirubin	5.0	1.3–20.0
						AST/ALT ratio ≥ 2	7.1	1.8–28.1

ICU: Intensive care unit; RRT: Renal replacement therapy; AST: Aspartate Aminotransferase; ALT: Alanine transaminase; LDH: Lactate dehydrogenase.

## Data Availability

Data are available from the Far North Queensland Human Research Ethics Committee. (contact via email fnq_hrec@health.qld.gov.au) for researchers who meet the criteria for access.

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
