# Peer review of "The Characteristics of Patients That Develop Severe Leptospirosis: A Scoping Review"

_pathogens, 2025, doi:10.3390/pathogens14121268_

Round 1

Reviewer 1 Report

Comments and Suggestions for Authors

General Comment:
     The authors reviewed severe cases of leptospirosis reported in the literature, collected before 1 March 2025, from Asia, the Americas, Europe, and Oceania. They found that the disease most commonly affects older individuals and is associated with hypotension, respiratory abnormalities, renal impairment, jaundice, and altered mental status. Simple clinical signs and basic laboratory tests—particularly complete blood count and assessment of kidney and liver function—demonstrate strong prognostic value across regions. Early consideration of these parameters may facilitate identification of high-risk patients and enable timely intervention, even in resource-limited settings, supporting effective management of this life-threatening disease.

    I greatly appreciate the authors’ work—this is a highly valuable review that makes an important contribution to understanding and managing leptospirosis.

    In my opinion, the review was conducted rigorously and systematically, with independent dual screening, consensus discussions, and NOS-based risk-of-bias assessment.

     I have only a few minor suggestions, though two points may deserve slightly more attention:

  1. Geographical classification: Turkey was classified as part of Europe (72 patients), although most of its territory lies in Asia (97%). It may be worth considering an alternative classification for Turkey to better reflect regional differences in the analysis.
  1. Pathogen-related factors: The authors might consider including an analysis of pathogen-related factors, particularly specific Leptospira serovars, as this could provide additional insight into regional differences in clinical presentation and disease severity.
  2. Figure captions: In the manuscript, figure captions are placed above the figures, whereas legends are usually positioned below the figures. It may be advisable to adjust this to align with common editorial practice.

Author Response

Reviewer 1

The authors reviewed severe cases of leptospirosis reported in the literature, collected before 1 March 2025, from Asia, the Americas, Europe, and Oceania. They found that the disease most commonly affects older individuals and is associated with hypotension, respiratory abnormalities, renal impairment, jaundice, and altered mental status. Simple clinical signs and basic laboratory tests—particularly complete blood count and assessment of kidney and liver function—demonstrate strong prognostic value across regions. Early consideration of these parameters may facilitate identification of high-risk patients and enable timely intervention, even in resource-limited settings, supporting effective management of this life-threatening disease.

    I greatly appreciate the authors’ work—this is a highly valuable review that makes an important contribution to understanding and managing leptospirosis.

    In my opinion, the review was conducted rigorously and systematically, with independent dual screening, consensus discussions, and NOS-based risk-of-bias assessment.

Response: We thank Reviewer 1 for the time that he/she has taken to review our manuscript and the very helpful suggestions for its enhancement. We are encouraged to hear that he/she feels that it is a highly valuable review that makes an important contribution to understanding and managing leptospirosis. Please find below a point-by-point response to his/her comments.

     I have only a few minor suggestions, though two points may deserve slightly more attention:

  1. Geographical classification: Turkey was classified as part of Europe (72 patients), although most of its territory lies in Asia (97%). It may be worth considering an alternative classification for Turkey to better reflect regional differences in the analysis.

Response: We thank the Reviewer for this suggestion. Geographical classification can be difficult; indeed, wars have been fought over this! We feel that Türkiye sits more neatly into Europe in our classification and indeed, Türkiye itself – which is partly in Europe – has European aspirations: Türkiye has been a member of NATO since 1952, applied to join the European Economic Community (EEC) in 1987 and in 1999 it was recognised as a candidate country for EU membership.

Of course, the pathogen knows nothing of geographical borders, but Türkiye is approximately 569 km from Bulgaria, 884 km from Ukraine and 1832 km from France, the three countries that represent the other studies. In contrast, Türkiye is approximately 7521 km from Thailand,  8,452 km from Malaysia, 8,856 km from the Philippines and 6,347 km from Sri Lanka, where most of the Asian studies were performed.

Our classification is necessarily imperfect – Réunion Island is part of France but is located in the Indian Ocean. However, we have been transparent on our presentation of the studies, and we don’t feel that the presentation of the data of these 72 patients in the table listing European series rather than Asian series has any impact on the findings or conclusions of the study.

  1. Pathogen-related factors:The authors might consider including an analysis of pathogen-related factors, particularly specific Leptospira serovars, as this could provide additional insight into regional differences in clinical presentation and disease severity.

Response: We thank the Reviewer for this suggestion, but detailed analysis of pathogen related factors was beyond the scope of our study, which had a focus instead (as the title of the paper suggests) on the characteristics of patients that develop severe leptospirosis.

The aim of the study was to identify the findings in the patients that were associated with the development of severe disease to facilitate the recognition of the high-risk patient, expediting the escalation of their care. We agree that further data are needed to define the independent contribution of pathogen related factors (including the degree of leptospiraemia, the serovar and virulence factors). We do discuss the role of serovars in our study (lines 356-362) emphasise this as a focus for future research in our discussion (lines 405-407).

However, globally, the vast majority of the patients who develop leptospirosis are living in rural or resource-limited settings where this is limited access to diagnostic support: pathogen related data including the serovar will not be available for several days, if at all. In these locations the clinical and simple laboratory findings will be of immediate practical utility to the attending clinician (and pragmatically speaking is the relevant endpoint of pathogen-related factors).

Indeed, one of the main findings of our study was that although serovars of leptospirosis undeniably vary geographically, the presence of hypotension, respiratory or renal involvement had prognostic utility in all regions.

  1. Figure captions: In the manuscript, figure captions are placed above the figures, whereas legends are usually positioned below the figures. It may be advisable to adjust this to align with common editorial practice.

Response: We thank the Reviewer for this suggestion. We have moved the figure captions below the figures as suggested. We are happy for the captions to be presented in any way that the Editorial staff feel appropriate.

Reviewer 2 Report

Comments and Suggestions for Authors

This review paper provides a comprehensive and useful overview of the signs that indicate a high risk of severe leptospirosis. The authors compiled numerous studies from various countries, which helps the reader understand how signs vary between regions. The structure of the paper is logical, and the flow from one section to the next is smooth. The scientific content is strong. However, some parts of the text are too lengthy, and certain ideas are repeated more than once. The tables contain a large amount of data, which makes them difficult to read quickly. The message of the paper is important, but it can be made more clear with better focus and shorter explanations.

The tables contain a large amount of information from the included studies. This is useful, but it also makes the tables heavy and difficult to read. It may help if the authors add brief summary lines after each table, so the reader can see the main message without having to read every row. Some studies lack confidence limits for the reported odds ratios, and these should be included if available.

The discussion covers all the important points, but it is quite lengthy and contains some repeated ideas. For example, the need for early care is mentioned several times in similar words. The section on risk scores, such as SPiRO and QuickLepto, is informative; however, the text could be condensed without losing meaning. It would also help if the authors more clearly connect each main idea in the discussion to the corresponding findings in the results.

Author Response

Reviewer 2

This review paper provides a comprehensive and useful overview of the signs that indicate a high risk of severe leptospirosis. The authors compiled numerous studies from various countries, which helps the reader understand how signs vary between regions. The structure of the paper is logical, and the flow from one section to the next is smooth. The scientific content is strong.

However, some parts of the text are too lengthy, and certain ideas are repeated more than once. The tables contain a large amount of data, which makes them difficult to read quickly. The message of the paper is important, but it can be made more clear with better focus and shorter explanations.

Response: We thank Reviewer 2 for the time that he/she has taken to review our manuscript and the very helpful suggestions for its improvement. We are happy to hear that he/she feels that the paper is logical, the flow from one section to the next is smooth and the scientific content is strong. Please find below a point-by-point response to his/her comments.

The tables contain a large amount of information from the included studies. This is useful, but it also makes the tables heavy and difficult to read. It may help if the authors add brief summary lines after each table, so the reader can see the main message without having to read every row. Some studies lack confidence limits for the reported odds ratios, and these should be included if available.

Response: We thank Reviewer 2 for this suggestion. The tables are merely for reference for the interested reader. We do, in fact, summarise the main findings of the studies, quite succinctly, in Figure 2 and the key findings are also summarised in the text of the results (lines 162-316). We have therefore not added in a further summary.

The confidence intervals were available in 26 of the 35 articles and we have added these into the table. We have also rounded all confidence intervals to one decimal point to harmonise the presentation of data and to improve legibility.

The discussion covers all the important points, but it is quite lengthy and contains some repeated ideas. For example, the need for early care is mentioned several times in similar words. The section on risk scores, such as SPiRO and QuickLepto, is informative; however, the text could be condensed without losing meaning. It would also help if the authors more clearly connect each main idea in the discussion to the corresponding findings in the results.

Response: We thank Reviewer 2 for these constructive and helpful suggestions. As the focus of the paper is identifying the findings that can facilitate the recognition of the high-risk patient to expedite the escalation of care, some repetition of this concept is unavoidable, however, in the revised manuscript we have removed a reference to early escalation of care in the paragraph preceding the conclusions.

We have also condensed the discussion on the SPiRO and QuickLepto scores as suggested, removing three lines which did not enhance the point we were making.

We have also added some text to the discussion to link the discussion and results as suggested, highlighting that, specifically, serum creatinine had significant prognostic utility while the bedside finding of oliguria also had value in settings with less laboratory support (lines 344-346)

Round 2

Reviewer 1 Report

Comments and Suggestions for Authors

Dear Authors,

     Thank you for your detailed responses to my previous comments. I appreciate the effort and clarifications provided. However, I would like to maintain my concerns regarding two specific points:

  1. Geographical classification of Türkiye

     I understand the authors’ rationale for classifying Türkiye as part of Europe, based on political and historical affiliations. Nevertheless, from an epidemiological perspective, this classification is not fully justified. Türkiye lies predominantly in Asia (approximately 97% of its territory) and shares ecological and epidemiological characteristics with neighboring Asian and Middle Eastern regions.

     Including Türkiye in the European dataset may introduce heterogeneity and could obscure meaningful regional differences, particularly in studies comparing clinical patterns across continents. While I acknowledge that any classification is inevitably imperfect, and that the authors have transparently reported the data, I believe that a classification reflecting geographical and epidemiological realities would provide a more accurate framework for interpretation.

  1. Pathogen-related factors

     I also appreciate the authors’ explanation that the study focused on patient-related characteristics and the practical limitations of pathogen-level data in resource-limited settings. Nevertheless, pathogen-related factors, including Leptospira serovars and virulence determinants, remain relevant for interpreting regional differences in disease presentation and severity.

     While clinical and basic laboratory parameters are of immediate value to frontline clinicians, a more comprehensive epidemiological analysis—even if limited or exploratory—could benefit from incorporating available serovar or virulence data. Several serovars are known to correlate with specific organ involvement and clinical patterns; overlooking these factors may limit the biological explanation for observed regional differences.

     Once again, I leave the decision regarding the inclusion or modification of pathogen-related elements to the Editor.

     In summary, while I respect the authors’ approach and recognize practical limitations, I respectfully maintain that:

  1. The classification of Türkiye as European may not fully reflect epidemiological realities, and
  2. Pathogen-related factors remain relevant for interpreting regional differences and could be incorporated or discussed more systematically.
